# Choosing the parameter of the Fermat distance: navigating geometry and noise

**Frédéric Chazal**                                              *frederic.chazal@inria.fr*
*Institut de Mathématique d'Orsay, Faculté des Sciences d'Orsay, Université Paris-Saclay, France*

**Laure Ferraris**                                               *laure.ferraris@inria.fr*
*Institut de Mathématique d'Orsay, Faculté des Sciences d'Orsay, Université Paris-Saclay, France*

**Pablo Groisman**                                               *pgroisma@dm.uba.ar*
*IMAS-CONICET, Dep. de Matemática, Fac. Cs. Exactas y Naturales, Universidad de Buenos Aires, Argentina*

**Matthieu Jonckheere,**                                         *matthieu.jonckheere@laas.fr,*
*LAAS-CNRS, Université de Toulouse, CNRS, Toulouse, France*

**Frédéric Pascal**                                              *frederic.pascal@centralesupelec.fr*
*Université Paris-Saclay, CNRS, CentraleSupélec, Laboratoire des Signaux et Systèmes, 91190, Gif-sur-Yvette, France*

**Facundo Sapienza**                                             *fsapienza@berkeley.edu*
*Department of Statistics, University of California, Berkeley, CA, USA*

**Reviewed on OpenReview:**

## Abstract

The Fermat distance has been recently established as a valuable tool for machine learning tasks when a natural distance is not directly available to the practitioner or to improve the results given by Euclidean distances by exploiting the geometrical and statistical properties of the dataset. This distance depends on a parameter $\alpha$ that significantly affects the performance of subsequent tasks. Ideally, the value of $\alpha$ should be large enough to navigate the geometric intricacies inherent to the problem. At the same time, it should remain restrained enough to avoid any deleterious effects stemming from noise during the distance estimation process. We study both theoretically and through simulations how to select this parameter.

***Keywords*** — Fermat distance, clustering, geometry, density, metric learning

## 1 Introduction

The Fermat distance and other density based distances have recently attracted interest in machine learning and shown usefulness in various application domains. For example, they have been used in topological data analysis to detect cancer fingerprints (4), in signal analysis to detect periodicity and anomalies in ECG and in the reconstruction of dynamical systems (9). The also have shown interesting potential in semi-supervised learning and in definition of notions of data depth (3). Exploiting both the geometric and statistical properties of datasets, the Fermat distance favor geodesic paths to go through high data density areas - by reducing distances between points there - and avoid low density areas - by increasing distances between points there - of the space in which they are defined. The influence of the density in the definition of the Fermat distance depends on a parameter $\alpha$ whose choice is critical: the performances of subsequent tasks, such as clustering, tend to improve as $\alpha$ increases while at the same time the estimation of the Fermat distance tends to become dramatically sensitive to noise. The overall objective of this paper

is to better understand the role and behavior of this parameter $\alpha$ in the estimation of the Fermat distance and its impact on clustering subsequent tasks.

Let $Q_n = \{q_1, \ldots, q_n\} \subset \mathbb{R}^D$ be independent random points with common density $f : \mathcal{M} \subset \mathbb{R}^D \mapsto \mathbb{R}_{\geq 0}$ supported on a smooth Riemannian manifold $\mathcal{M}$ embedded in $\mathbb{R}^D$. We assume that $f$ is a density with respect to the volume form on $\mathcal{M}$. Typically, $d := \dim(\mathcal{M}) < D$, but it can take any value $1 \leq d \leq D$.

We first recall the definition of the *sample Fermat distance* and the *macroscopic Fermat distance* (14; 11; 18).

**Definition 1.1** (Sample Fermat distance). For $\alpha \geq 1$ and $x, y \in \mathbb{R}^D$ we define,

$$D_{Q_n,\alpha}(x,y) = \inf \left\{ \sum_{j=1}^{k-1} |q_{j+1} - q_j|^\alpha : (q_1, \ldots, q_k) \text{ is a path from } x \text{ to } y, \ k \geq 1 \right\}, \tag{1.1}$$

where $|\cdot|$ denotes the $D$-dimensional Euclidean norm, $q_1$ and $q_k$ are the nearest neighbors in $Q_n$ of $x$ and $y$, respectively.

**Definition 1.2** (Macroscopic Fermat distance). For $\alpha \geq 1$ and $x, y \in \mathcal{M}$,

$$D_\alpha^d(x,y) = \inf_{\gamma \in \Gamma_{x,y}} \int_\gamma \frac{1}{f^{(\alpha-1)/d}}, \tag{1.2}$$

where $\Gamma_{x,y}$ is the set of rectifiable paths $\gamma : [0,1] \to \mathcal{M}$ such that $\gamma(0) = x$, $\gamma(1) = y$.

These distances have been used in several situations and proved to be useful in a variety of learning tasks (18; 16; 9; 11; 26; 5; 17). Under different assumptions on $\mathcal{M}$ and $f$, it has been shown that when $\alpha > 1$, as $n \to \infty$,

$$n^{(\alpha-1)/d} D_{Q_n,\alpha}(x,y) \longrightarrow \mu(\alpha, d) D_\alpha^d(x,y), \qquad \text{a.s.}, \tag{1.3}$$

where $\mu(\alpha, d)$ is a positive constant depending only on the intrinsic dimension $d$ of $\mathcal{M}$ and the parameter $\alpha$ (11; 9; 14).

It is worth noting that in the works (16; 18), a different definition of Fermat distance is introduced, specifically expressed as

$$\bar{D}_{Q_n,\alpha}(x,y) = D_{Q_n,\alpha}(x,y)^{1/\alpha}. \tag{1.4}$$

This alternative definition carries implications for practical implementations, each accompanied by its advantages and limitations, which must be considered when using it along with specific subsequent machine learning tasks.

Our study sheds light on some key points about the use of the Fermat distance, as given in Definition 1.2, for both clustering and classification tasks.

**Contributions.** Firstly, we demonstrate the existence of a critical value $\alpha_0$, dependent on the underlying distribution's geometric parameters and its intrinsic dimensionality. For values $\alpha > \alpha_0$, both classification and clustering tasks become feasible for large sample sizes, which implies that the derivation of a meaningful distance measurement hinges upon $\alpha$ exceeding this pivotal threshold. Under regularity assumptions, we show that $\alpha_0$ scales linearly with the dimension. Contrary to this point, we also highlight that using large values of $\alpha$ leads to drawbacks due to increased variability in the sample Fermat distance. This effect has a negative impact on the distance estimation for finite samples. We prove that the variance of the sample Fermat distance scales exponentially with $\alpha$ in dimension one, and we obtain exponential bounds in terms of $\alpha/d$ in higher dimensions. We conclude that a sensible choice for $\alpha$ is crucial and provide guidelines to perform this choice.

To illustrate our findings, we conduct experiments on synthetic datasets and observe that there is a practical critical window of values of $\alpha$ where the performance is optimal. Although the definitions and results presented in this study are relevant to both the supervised context of classification and the unsupervised tasks of clustering, our subsequent analysis in the following sections will be framed within the context of clustering analysis.

## 2 Population clusters and a lower bound on $\alpha$

We start with a definition of clusters that extends the one given by the level sets of a density.

**Definition 2.1** (Clusters). For a measure with density $f$ (with respect to the volume form) on the Riemannian manifold $\mathcal{M} \subset \mathbb{R}^D$ and a family of sets $\mathcal{C} := (C_i)_{i=1}^m$, $C_i \subset \mathbb{R}^D$, we say that $\mathcal{C}$ are $\alpha-$clusters of $f$ if there exists $\epsilon > 0$ such that

$$D_\alpha^d(x, y) \leq D_\alpha^d(x, y') - \epsilon, \text{ for all } 1 \leq i \leq m, j \neq i, x, y \in C_i, y' \in C_j. \tag{2.1}$$

If this holds for some $\alpha \geq 1$, we will say that the clustering problem $(\mathcal{C}, f)$ is $\alpha-$feasible or just feasible.

Note that this definition of clusters aligns with clustering methods that find clusters by comparing intra-cluster distances with inter-cluster distances. In practice, this is usually done by finding centers or centroids $c_1, \ldots, c_m$ (*e.g.*, $K-$medoids or $K-$means techniques) such that,

$$D_\alpha^d(x, c_i) \leq D_\alpha^d(x, c_j) - \epsilon, \text{ for all } 1 \leq i \leq m, j \neq i, x \in C_i. \tag{2.2}$$

This last notion is the one that is used to find the empirical centroids (*i.e.*, estimators of $c_1, \ldots, c_m$) and then the clusters. We could adopt this definition and state our results regarding this property, but we prefer to use (2.1) to avoid dealing with centroid estimations and their consistency properties. We remark that all the statements and definitions below can be rephrased to fit the notion of clusters that involves centroids.

If a macroscopic clustering problem is feasible, we define the *critical parameter* as the least $\alpha_0 \geq 1$ for which (2.1) holds for every $\alpha > \alpha_0$. More precisely,

$$\alpha_0 = \inf\{\alpha' \geq 1 : \mathcal{C} \text{ are } \alpha-\text{clusters of } f \text{ for all } \alpha > \alpha'\}. \tag{2.3}$$

We will demonstrate that using the Fermat distance in our definition enables us to address various natural scenarios characterized by clusters effectively. This also generalizes other notions of clusters based on level sets of $f$ (see Proposition 2.2 below). We will also show the existence of a finite critical parameter $\alpha_0$ under mild conditions. Observe that the clusters of $f$ need not be uniquely defined.

### 2.1 Preliminaries

We first introduce notations used in the sequel. Given a set $C \in \mathbb{R}^D$, *the distance function* to $C$, denoted by $d_C$, is defined by

$$d_C(x) = \inf_{y \in C} |x - y|. \tag{2.4}$$

Notice that since $\overline{C} = \left\{ x \in \mathbb{R}^D \ : \ d_C(x) = 0 \right\}$, the distance function $d_C$ fully characterize $C$ as soon as $C$ is closed.

We call *r-offset* of $C$, denoted by $C^r$, the set of points at distance at most $r$ of $C$, or equivalently the sublevel set defined by

$$C^r = \left\{ x \in \mathbb{R}^D \, | \, d_C(x) \leq r \right\}. \tag{2.5}$$

Finally, we define the *reach* of $C$, denoted as $\mathrm{rch}(C)$ as the largest value $r > 0$ such that for all $y \in C^r$, there exists a unique $x \in C$ such that $|x - y| = d_C(y)$. The reach of a set quantifies how far a compact set $C$ is from being convex. Notice that $\mathrm{rch}(C)$ is small if either $C$ is not smooth or if $C$ is close to being self-intersecting. On the other hand, if $C$ is convex, then $\mathrm{rch}(C) = \infty$.

Consider two disjoint, connected and compact sets $A, B \subset \mathbb{R}^D$ such that $C = A \cup B$. We denote the shortest Euclidean distance between $A$ and $B$ by

$$\mathrm{dist}(A, B) = \inf \left\{ |a - b| \ : \ a \in A, \, b \in B \right\}. \tag{2.6}$$

If $\mathrm{dist}(A, B) = \delta > 0$ and if we call $p \in \mathbb{R}^D \setminus C$ the middle point of a shortest straight line from $A$ to $B$, then the projection of $p$ onto $C$ is not unique as it has at least one candidate on $A$ and one on $B$. We can then state that $\mathrm{rch}(C) \leq \delta/2$. Moreover, if $A$ and $B$ are convex, $\mathrm{rch}(C) = \delta/2$. The interested reader can refer to (1; 8; 6) for a more detailed introduction to these concepts.

## 2.2 Existence of a critical parameter

We can now state the existence of a critical parameter $\alpha_0$.

**Proposition 2.2.** *Let $\mathcal{C} = (C_i)_{i=1}^m$ be a family of compact and connected sets such that $C = \bigcup_{i=1}^m C_i$ has positive reach $\tau > 0$. Suppose that $f(x) \geq a_1$ for all $x \in C$, and $f(x) \leq a_0$ for all $x \notin C$, with $a_1 > a_0$. Then, there exists a positive constant $\beta_0 \geq 0$ depending on the diameters of the $C_i$'s, the reach of $C$ and the intrinsic dimension $d$ of $C$ such that for all*

$$\alpha > 1 + d\, \frac{\beta_0}{\log(a_1/a_0)}\,, \tag{2.7}$$

*the macroscopic clustering problem $(\mathcal{C}, f)$ is $\alpha-$feasible. If the length of geodesics in $C$ is uniformly bounded, the constant $\beta_0$ can be taken independent of $d$ (and hence, the bound from below for the critical parameter is linear in $d$).*

*Proof.* The proof of this proposition is postponed to Appendix A. $\qquad\square$

**Remark 2.3.** As pointed out before, the reach encodes the geometric complexity of the problem. A small reach is associated with a more complex problem. This is present in the previous result in the dependence of the constant $\beta_0$ on the reach. The constant $\beta_0$ can be computed explicitly, and in fact, an explicit formula is given in the course of the proof of the proposition. When $C$ is such that the geodesic path lengths are not uniformly upper bounded, the parameter $\beta_0$ turns out to scale proportional to $d$, giving that the bound from below for $\alpha_0$ scales as $d^2$.

**Remark 2.4.** The hypothesis of Proposition 2.2 forces $f$ to be discontinuous (otherwise, we necessary have $a_0 = a_1$). We prefer to first deal with this restrictive situation to simplify the exposition, but in fact, this restriction can be removed to cover the case when $f$ is a continuous density by defining $a_0$ more carefully. In that case, we choose $a_0 < a_1$ in such a way that there is a region around each cluster that separates the level sets $f^{-1}(a_0)$ and $f^{-1}(a_1)$. It is worth observing the behavior of the lower bound as $a_0 \to 0$ and as $a_0 \to a_1$.

**Proposition 2.5.** *Let $\mathcal{C} = (C_i)_{1 \leq i \leq m}$ and $C = \bigcup C_i$ as before. Suppose that $f \geq a_1$ in $C$, and let $a_0$ be such that $0 < \eta = \inf\left\{|s - t| \,:\, s \in f^{-1}([0; a_0]), \, t \in f^{-1}([a_1; \infty])\right\} < \tau = rch(C)$. Then, there exists $c > 0$ such that for every $0 < r < \tau - \eta$*

$$\alpha > 1 + d\, \frac{\log\left(\frac{r}{\tau - (r+\eta)}(cr^{-d} \vee 1)\right)}{\log(a_1/a_0)}, \tag{2.8}$$

*the macroscopic clustering problem is $\alpha-$feasible. If the length of geodesics in $C$ is uniformly bounded, we can take $\alpha$ as in (2.7) with the constant $\beta_0$ independent of $d$ (and hence the bound from below for the critical parameter is linear in $d$).*

*Proof.* The proof is postponed to Appendix A. $\qquad\square$

# 3 Empirical clusters

In this section, we leverage the concepts established in the preceding section to address the clustering quandary in the context of finite samples. Our goal is to ascertain the specific conditions under which a clustering problem can be solved solely by harnessing the properties of the Fermat distance.

**Definition 3.1** (Clustering). Given a family of clusters $(C_i)_{i=1}^m \subset \mathbb{R}^D$ and points $Q_n = \{q_1, \ldots, q_n\} \subset \mathbb{R}^D$, we say that the empirical clustering problem is $\alpha-$feasible for $Q_n$ if there exists $\epsilon > 0$ such that

$$n^{(\alpha-1)/d} D_{Q_n,\alpha}(x, y) \leq n^{(\alpha-1)/d} D_{Q_n,\alpha}(x, y') - \epsilon, \tag{3.1}$$

for all $1 \leq i \leq m$, $x, y \in Q_n \cap C_i$, and $y' \in Q_n \cap C_j$, $j \neq i$. When $Q_n$ is random, we call $F(\alpha, \epsilon, n) = F(\alpha, \epsilon, Q_n, C_1, \ldots, C_m)$ the event under which the samples are such that (3.1) holds.

The idea here is that a simple classification rule using Fermat distance should be sufficient to reach a perfect clustering. We can now state the main result of this section, which relates the macroscopic problem to the microscopic one.

**Proposition 3.2.** *Let $\mathcal{C} = (C_i)_{1 \leq i \leq m}$ be a family of disjoint, compact and connected sets, and $C = \bigcup_{i=1}^{m} C_i$. Assume $(\mathcal{C}, f)$ is $\alpha-$feasible for $\alpha$ large enough. Assume further that $C$ is a closed $d-$dimensional Riemannian manifold embedded in $\mathbb{R}^D$ and $f$ is a smooth probability density function with $\liminf_{\mathcal{M}} f(x) > 0$.*

*Then there exist $\epsilon > 0$, constants $\delta, c, \gamma > 0$ and an integer $\bar{n}$ such that for $n \geq \bar{n}$, we have for $\alpha > \alpha_0$,*

$$\mathbb{P}(F(\alpha, \epsilon, n)^c) \leq e^{-cn^\gamma}.$$

*Conversely, for every $\delta > 0$ there is $\alpha_0 - \delta < \alpha < \alpha_0$ such that for every $n \geq \bar{n}$,*

$$\mathbb{P}(F(\alpha, \epsilon, n)^c) \geq 1 - e^{-cn^\gamma}.$$

*Proof.* The proof is given in Appendix A ◻

**Remark 3.3.** Following (9), given $\alpha$ and $d$, the constant $\gamma$ can be chosen in the open interval $(0, \alpha(2\alpha + d))$. It is an open problem to determine $\bar{n}$ as a function of $\alpha$ and $d$.

### 3.1 Example with piecewise constant densities (clutter)

In $\mathbb{R}^d$, we denote the $d-$dimensional ball centered in the origin 0 with radius $r$ by $\mathbb{B}_d(r)$, and the volume of $\mathbb{B}_d(1)$ by $\omega_d$. Consider two $d-$dimensional ring $C_1, C_2$ centered at 0 contained in an hypercube $K$: $C_1 = \mathbb{B}_d(\frac{5}{4}) \smallsetminus \mathbb{B}_d(1)$, $C_2 = \mathbb{B}_d(\frac{10}{4}) \smallsetminus \mathbb{B}_d(\frac{9}{4})$, $K = [-3, 3]^d$. Let $P_C$ be the uniform measure supported on $C = C_1 \cup C_2$, and $P_K$ be the uniform measure supported on $K$. Suppose $Q_n = \{q_1, \ldots, q_n\}$ is a random sample generated from a mixture of $P_C$ and $P_K$,

$$X_1, \ldots, X_n \equiv P_{cl} \sim \lambda P_C + (1 - \lambda)P_K, \tag{3.2}$$

with $\lambda \in [0, 1]$ being a proportion parameter that quantifies the signal-to-noise ratio in the sample (see Figure 1a).

This model is usually called *clutter noise* in the literature. We are observing data points near or in $C$, but we aim at finding a bi-partition of $K$ such that $C_1$ and $C_2$ are strictly contained in two different clusters. We plug the sample Fermat distance in the $K-$medoids algorithm, that we denote Fermat $K-$medoids (see Section 5 for more details). We are interested in the relation between the parameter $\alpha$ and the predictions for any point in $C_1 \cup C_2$.

Notice that the distribution $P_{cl}$ admits a density $f$ with respect to the Lebesgue measure,

$$f(x) = \frac{\lambda}{|C_1| + |C_2|}\left(\mathbf{1}_{C_1}(x) + \mathbf{1}_{C_2}(x)\right) + (1 - \lambda)\frac{1}{|K|}\mathbf{1}_K(x), \tag{3.3}$$

where $|C_1|$, $|C_2|$ and $|K|$ denote the volume of $C_1$, $C_2$ and $K$, respectively.

Our goal now is to find an explicit bound for the critical parameter. The argument follows the same lines as the proof of Proposition 2.2 but with explicit quantities. The idea is that it is sufficient to set $\alpha$ such that the Fermat cost of the shortest straight line between $C_1$ and $C_2$ and outside $C_1 \cup C_2$, is always greater than any Fermat distance between two points inside the same cluster. We need to compute the longest geodesic distance (with respect to the Euclidean norm) lying in $C$ denoted by $L$. This model's explicit upper bound is $L \leq 5\pi/2$. In the proof of Proposition 2.2, the upper bound on $L$ is computed for a general case and depends on the $r$-covering number of $C$, with $0 < r < \text{rch}(C)$. We have, $\text{rch}(C) = \text{dist}(C_1, C_2)/2 = 1/2$, $a_1 = \lambda/(|C_1| + |C_2|)$, and $a_0 = 1 - \lambda/|K|$.

Any continuous path that intersects both $C_1$ and $C_2$ has an Euclidean length of at least $\text{dist}(C_1, C_2) = 1$. Then, the Fermat distance of any path lying in two distinct clusters is at least $a_0^{(1-\alpha)/d}$. It suffices to choose $\alpha$ such that the Fermat cost between $C_1$ and $C_2$ is always greater than

$$a_0^{(1-\alpha)/d} > L\, a_1^{(1-\alpha)/d}, \qquad \alpha > 1 + d\,\frac{\log(L)}{\log(a_1/a_0)}. \tag{3.4}$$

We have obtained the upper bound

$$\alpha_0 \leq 1 + d\frac{\log\left(\frac{5\pi}{2}\right)}{\log(a_1/a_0)}, \tag{3.5}$$

meaning that $(C_1, C_2)$ are $\alpha-$clusters for every $\alpha \geq \alpha_0$. Figure 1 shows the result of applying $K-$medoids for different choices of $\alpha$ and different values of $\lambda$ with sample size $n$. We observe that we can achieve clustering above this threshold for the theoretical values of $\alpha_0$ found using (3.5).

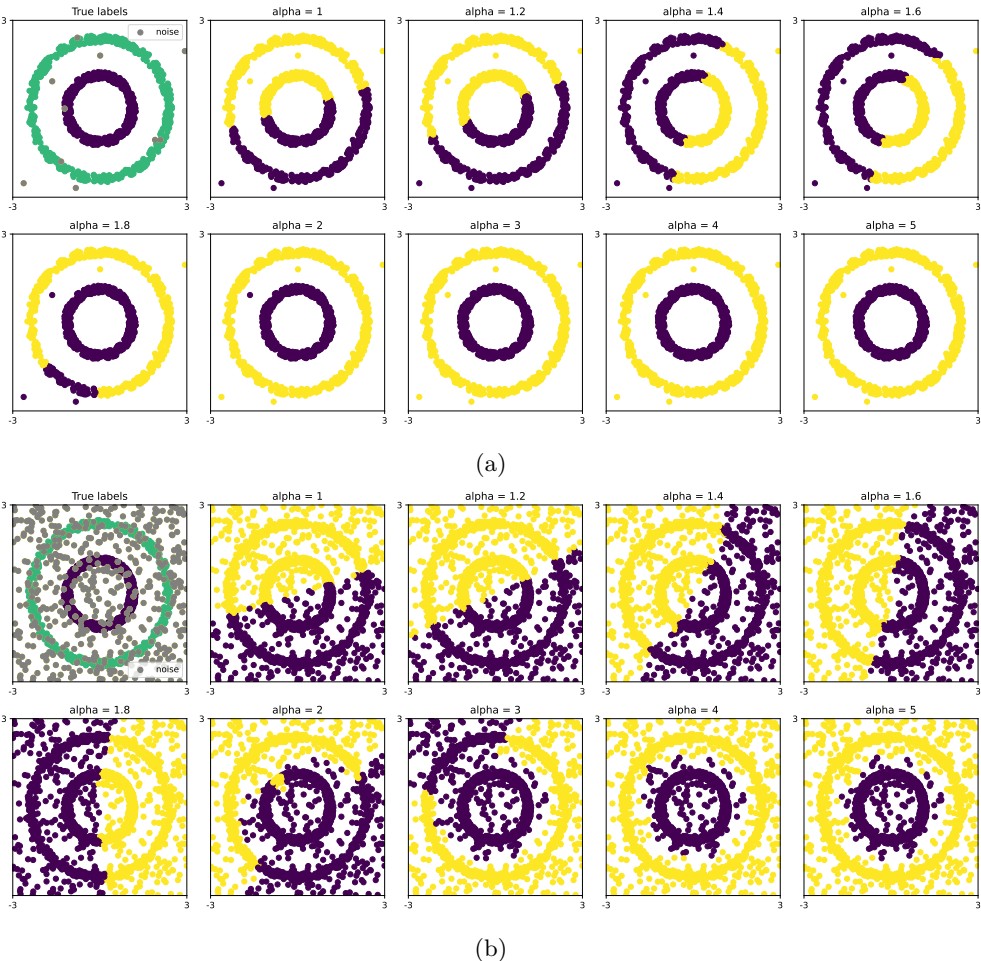

Figure 1: An example of predictions of the Fermat $K-$medoids algorithm on data with common density $f$ for (A) $\lambda = 0.99$, $n = 1000$ and (B) $\lambda = 2/3$, $n = 1500$. The theoretical values for the critical value of $\alpha$ based on (3.5) give $\alpha_0 \le 1.45$ and $\alpha_0 \le 2.2$, respectively, meaning that above those values we are able to perform an efficient clustering with high probability for large $n$.

Remark that $\alpha_0$ is the theoretical critical parameter. In practice, we use the sample Fermat distance, so the threshold fluctuates (see Proposition 3.2). This synthetic example illustrates that the parameter $\alpha$ calibrates the sensitivity of the distance to both the geometry and density of the data. Note that when $\alpha = 1$, the Fermat distance boils down to the Euclidean distance, and predictions result from the classical $K-$medoids algorithm. By pushing up the $\alpha$ value, the algorithm captures the structure of both rings despite the noise and performs a partition that perfectly identifies $C_1$ and $C_2$.

## 4   Variability as a function of $\alpha$

When opting for a large value of $\alpha$ to get a distance that aptly captures the intricacies of the geometry and the data distribution, it is essential to recognize that this choice can introduce two noteworthy challenges.

1. *Heightened variability of Sample Fermat Distance.* Choosing a large value of $\alpha$ can lead to amplified variability within the sample Fermat distance.

2. *Floating point arithmetic.* There is the concern of numerical inaccuracies arising due to the utilization of larger $\alpha$ that lead to the manipulation of numbers that differ in orders of magnitude.

This section delves into the first of these issues, furnishing preliminary insights into the exploration of variance within the sample Fermat distance statistic.

Deriving exact characterizations of the variance of $n^{(\alpha-1)/d} D_{Q_n,\alpha}(x, y)$ in dimensions larger than one have been shown to be challenging (2; 13) and is still an open problem in the context of First Passage Percolation. The one-dimensional case is much more tractable, but the computations are not obvious even in this case.

Since we are interested in comparing the variability of the Fermat distance for different values of $\alpha$, we need a measure that does not depend on the scale of the statistic as $\alpha$ varies. We consider the coefficient of variation of $D_{Q_n,\alpha}^d$,

$$\sqrt{\frac{\mathrm{Var}\left(D_{Q_n,\alpha}^d\right)}{\mathbb{E}[D_{Q_n,\alpha}^d]^2}}, \tag{4.1}$$

when $\alpha$ varies and $n$ is fixed. In particular, we aim to understand the effect of the parameter $\alpha$ on the variation, giving an idea of a threshold which we might better set $\alpha$ below in practice.

### 4.1 One-dimensional case

For the one-dimensional case, we can explicitly calculate the spacings between consecutive points in the optimal path. In that case, we are going to consider a sample $Q_n = \{q_1, q_2, \ldots, q_n\}$ of uniform independent points in $[0, 1]$ and study the microscopic Fermat distance. In this simple case, it is given by,

$$D_{Q_n,\alpha}^{d=1} = \sum_{i=0}^{n} |q^{(i+1)} - q^{(i)}|^\alpha =: \sum_{i=0}^{n} \Delta_i^\alpha \tag{4.2}$$

with $q^{(1)}, \ldots, q^{(n)}$ the order statistics of $Q_n$, $q^{(0)} = 0$ and $q^{(n+1)} = 1$.

**Proposition 4.1.** *The expectation and variance of $D_{Q_n,\alpha}^{d=1}$ for the uniform case verify*

$$\lim_{n \to \infty} \mathbb{E}[n^{\alpha-1} D_{Q_n,\alpha}^{d=1}] = \Gamma(\alpha + 1),$$

$$\lim_{n \to \infty} n \, \mathrm{Var}[n^{\alpha-1} D_{Q_n,\alpha}^{d=1}] = \Gamma(2\alpha + 1) - (\alpha^2 + 1)\Gamma(\alpha + 1)^2$$

*Proof.* The proof is provided in the Appendix A. □

This result shows how the variability of the Fermat distance increases exponentially with the value of $\alpha$. If we consider the coefficient of variation given by

$$\sqrt{\frac{\mathrm{Var}(D_{Q_n,\alpha}^{d=1})}{\mathbb{E}[D_{Q_n,\alpha}^{d=1}]^2}} \underset{n \to \infty}{\smile} \frac{1}{\sqrt{n}} \sqrt{\frac{\Gamma(2\alpha+1) - (\alpha^2+1)\Gamma(\alpha+1)^2}{\Gamma(\alpha+1)^2}} \tag{4.3}$$

this scales exponentially as a function of $\alpha$, given large values even for small values of $\alpha$. Notice that this result further implies that the constant $\mu(\alpha, d)$ involved in Equation (1.3) satisfies $\mu(\alpha, 1) = \Gamma(\alpha + 1)$.

### 4.2 Higher dimensions

In higher dimensions, as already mentioned, the asymptotic behavior in $n$ of the variance of Fermat distance is an open problem even for Poisson point processes in Euclidean space. Similarly, we cannot prove a sufficiently informative bound on the variance of the Fermat distance in terms of the parameter $\alpha$. This is due to the difficulty in finding quantitative lower bounds for the Fermat distance.

However, we conjecture that the coefficient of variation scales exponentially in $\theta = \alpha/d$. This suggests that we should choose $\alpha$ in order to guarantee that $\alpha/d$ is not too large. We now prove a bound for the moments of the Fermat distance for an intensity one Poisson process in Euclidean space, showing that moments of order $k$ are at most of order $\Gamma(k\theta + 1)$.

**Proposition 4.2.** *Let $Q_n$ be a Poisson process with intensity $n$ in the hypercube of $\mathbb{R}^d$. Then we have,*

$$\mathbb{E}\left(\left(n^{\frac{\alpha-1}{d}}D^d_{Q_n,\alpha}(0,e_1)\right)^k\right) \leq 2e\,d^{\theta+\alpha/2}\theta^\theta(1+\Gamma(k\theta+1)).$$

*Proof.* The proof is provided in the Appendix A. □

## 5 Experiments

We consider here a series of clustering experiments involving the Fermat distance for different choices of $\alpha$. We are going to evaluate the trade-off between small values of $\alpha$ for which clustering is not feasible (Section 2)and large values of $\alpha$ where finite sampling effects distort the distance (Section 4).

### 5.1 Fermat $K-$medoids

We use the $K-$medoids algorithm for clustering (15; 10), a robust version of $K-$means in which the centroids of each cluster are forced to be a point in the set $Q_n$. An advantage of $K-$medoids is that different notions of distances can be used, leading to different partitions. The algorithm consists of alternating updates in the estimated cluster centroids $\hat{c}_i$ and cluster $\hat{C}_i \subset Q_n$ until the cluster assignment does not change. These updates are sequentially given by

$$\hat{c}_i \leftarrow \arg\min_{c \in C_i} \sum_{x \in \hat{C}_i} D_{Q_n,\alpha}(x,c), \tag{5.1}$$

$$\hat{C}_i \leftarrow \left\{x \in Q_n : D_{Q_n,\alpha}(\hat{c}_i,x) \leq D_{Q_n,\alpha}(\hat{c}_j,x)\,\forall j \neq i\right\}. \tag{5.2}$$

Note that different values of $\alpha$ give different partitions. If $\alpha = 1$, the sample Fermat distance boils down to the Euclidean distance, and both Fermat $K-$medoids and classical $K-$medoids coincide. For $\alpha > 1$, clusters are not necessarily convex.

### 5.2 Synthetic data

We evaluate the performance of the Fermat $K$-medoids clustering algorithm when analysing the Swiss-roll generated dataset. We consider four clusters sampled from a Gaussian distribution in two dimensions with different means and then mapped to three dimensions using the map $(x,y) \mapsto (x\cos(\omega x), ay, x\sin(\omega x))$ for $a = 3$, $\omega = 15$. We consider a total of $n = 1000$ random samples, equally split between four different clusters. This generates the dataset illustrated in Figure 2. When evaluating the performance of Fermat $K$-medoids using adjusted mutual information, adjusted rand index, accuracy, and F1 score (19; 27; 20), we observe that smaller values of $\alpha$ result in poor performances for all performance metrics. As we increase the value of $\alpha$, the performance increases until the performance decays again for large values of $\alpha$, having an optimal value at some middle range of values for $\alpha$. We also observe large variances in performance for large values of $\alpha$, resulting in decaying median performances over new samplings. We compare the performance using $K$-medoids but with the distance obtained using Isomap (25) and C-Isomap (24).

These experiments show that there is an optimal range of values of $\alpha$ that results in better clustering performance. This illustrates the point made in previous sections, showing that both small and large values of $\alpha$ result in poor performances.

### 5.3 Clustering MNIST

We now consider an example that inhabits a higher-dimensional and inherently more realistic realm: the digits 3 and 8 extracted from the MNIST dataset. Our approach begins by subjecting the data to preprocessing through PCA, resulting in a reduced-dimensional representation of dimension 40. We then cluster this representation using $K-$medoids with the Fermat distance. We compute the mean adjusted mutual information (AMI) and compare it to the performances of $K-$means with Euclidean distance and a robust EM procedure (21; 22) recognized for its consistently good performance as a fully unsupervised method.

The findings are consolidated in Figure 3. In this instance, the clear presence of an advantageous parameter range becomes apparent. Notably, the remarkable advantage gained from operating within this window, compared to

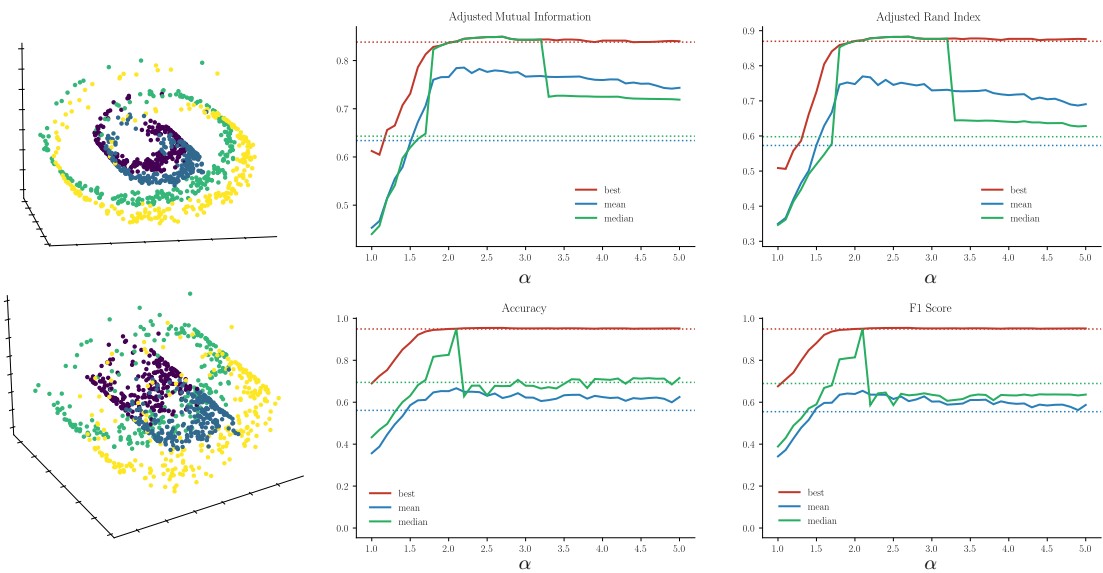

Figure 2: Different clustering performance indices for the Swiss Roll dataset. Solid lines correspond to the mean, median, and best performance archived by Fermat $K$-medoids over different samplings of the dataset. Dashed lines correspond to the performance obtained by the best output between Isomap and C-Isomap. This experiment has been presented previously in ICLR 2018 Workshop Track (23).

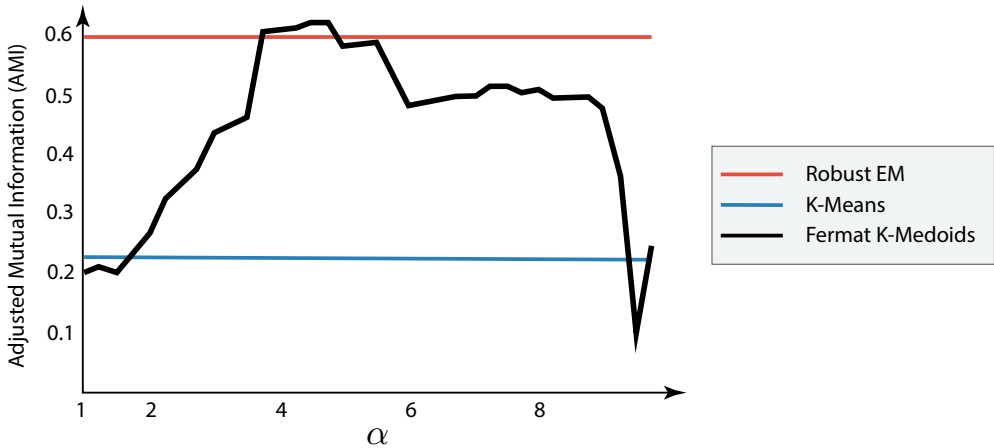

Figure 3: Performance of Fermat $K-$medoids algorithm compared to standard $K-$means and Robust expectation-maximization. Simulations credit to Alfredo Umfurer.

employing the Euclidean distance, is striking. Furthermore, it is remarkable that even a relatively straightforward algorithm like $K-$medoids can yield performance akin to significantly more computationally intricate methods when paired with the appropriate distance parameter.

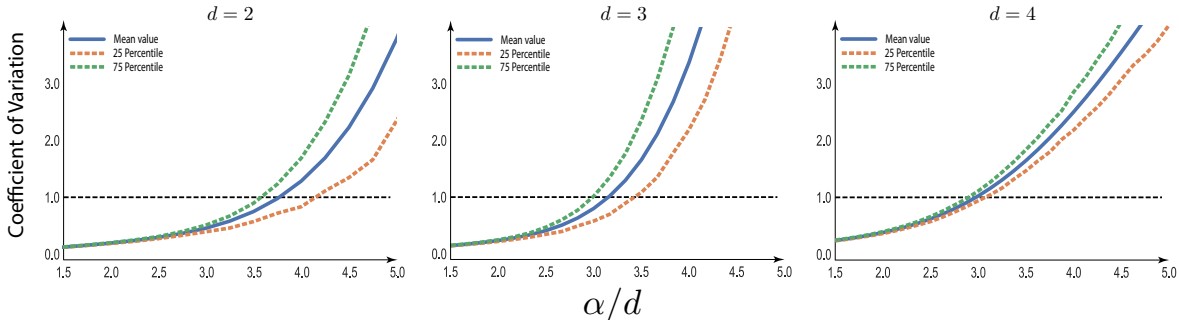

Figure 4: Coefficient of variation for the uniform distributions as a function of $\alpha/d$ for different ambient dimensions $d = 2, 3, 4$.

### 5.4 Coefficient of variation.

We conclude this section by conducting a numerical exploration of the influence of noise on the coefficient of variation of the Fermat distance. We plot in Figure 4 the coefficient of variation defined in 4.1 in the case of a uniform distribution on $[0, 1]^d$ as a function of $\alpha/d$. As conjectured, we observe an exponential behaviour in the parameter $\alpha/d$, demonstrated in dimension one and conjectured in higher dimensions.

### Acknowledgments

We acknowledge the support of the STIC AMSUD Project LAGOON for Pablo Groisman and Matthieu Jonckheere as well as the DATAIA support for Facundo Sapienza and Laure Ferraris.

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

## A  Proofs

*Proof of Proposition 2.2.* We prove that there exists $\alpha_0 \geq 1$ such that for all $\alpha \geq \alpha_0$ there is $\epsilon > 0$ such that for $1 \leq i \leq m$ and $x, y \in C_i$,

$$D_\alpha^d(x, y) \leq D_\alpha^d(x, y') - \epsilon, \quad \text{for all } y' \in C_j, \quad j \neq i.$$

First, we bound from below the Fermat distance between two points in two different sets. By assumption, $\mathrm{rch}(C) = \tau > 0$. Since $\tau \leq \min_{i \neq j} \mathrm{dist}(C_i, C_j)/2$, for all $x \notin C$, $f(x) \leq a_0$ and any path from $C_i$ to $C_j$ has measure at least $2\tau$ in the complement of $C$ when $i \neq j$, we have for $x \in C_i$, $y' \in C_j$,

$$D_\alpha^d(x, y') \geq a_0^{(1-\alpha)/d} 2\tau. \tag{A.1}$$

Second, we bound from above the Fermat distance between two points $x$ and $y$ in the same cluster $C_i$. Since $C$ is compact, we can consider a finite covering by $N_r$ $D$-dimensional balls, $\mathbb{B}_D$, with radius $r < \tau$, and centers $\mathcal{V} = \{v_1, \ldots, v_{N_r}\} \subset C$, *i.e.*,

$$C \subset \bigcup_{l=1}^{N_r} \mathbb{B}_D(v_l, r) \subset C^r.$$

We are going to build a path from $x$ to $y$ inside $C_i^r \subset C^r$ and project it onto $C$. By construction, we know that the path will be projected onto $C_i$. Let $\mathcal{V}_i = \{\mathcal{V} \cap C_i\}$. Observe that $(\mathbb{B}_D(v_l, r) \colon v_l \in \mathcal{V}_i)$ is a covering of $C_i$ by $\mathrm{card}(\mathcal{V}_i) = N_r(C_i) < N_r$ balls. Consider the weighted graph $\mathbb{G}_i$ with vertex set $\mathcal{V}_i$ in which two vertices are connected if and only if their respective balls intersect. The weight of an edge is given by the Euclidean distance between the two points. By construction, this graph is connected, and the weight of each edge is at most $2r$. We connect $x$ to the center $v(x) \in \mathcal{V}_i$ of a ball containing it. We have $|x - v(x)| \leq r$. Similarly, we connect $y$ to a $v(y) \in \mathcal{V}_i$ such that $|y - v(y)| \leq r$.

By definition, the shortest path from $x$ to $y$ inside $\mathbb{G}_i$, denoted by $\gamma$, cannot go through the same vertex more than once. Then the Euclidean length of $\gamma$, denoted by $|\gamma|$ verifies

$$|\gamma| \leq N_r(C_i)\, 2r \leq N_r\, 2r.$$

As $\gamma \subset C^r$ and $r < \tau$, the projection of $\gamma$ onto $C$, $\pi_C(\gamma) = \gamma_i$, is well defined. Moreover $\pi_C(\gamma) \subset C_i$. The projection map is Lipschitz, with Lipschitz constant $\dfrac{\tau}{\tau - r}$ ((8, Theorem 4.8)). Then we have,

$$|\gamma_i| = |\pi_C(\bar{\gamma})| \leq \frac{\tau}{\tau - r}\, N_r\, 2r.$$

The path $\gamma_i$, from $x$ to $y$, is contained in $C_i \subset C$ and for all $x \in C_i$, $f(x) \geq a_1$. Then,

$$D_\alpha^d(x, y) \leq \int_{\gamma_i} \frac{1}{f^{(\alpha-1)/d}} \leq a_1^{(1-\alpha)/d} \frac{\tau}{\tau - r}\, N_r\, 2r. \tag{A.2}$$

Next, we calibrate $\alpha$ to ensure the clustering feasibility, *i.e.*, one get,

$$a_0^{(1-\alpha)/d} 2\tau > a_1^{(1-\alpha)/d} \frac{\tau}{\tau - r}\, N_r\, 2r, \tag{A.3}$$

which is equivalent to,

$$\alpha > 1 + d\, \frac{\log\left(\dfrac{r}{\tau - r} N_r\right)}{\log(a_1/a_0)}.$$

Now, suppose that $N_r$ is optimal, *i.e.*, it is the smallest number of balls $\mathbb{B}_D(v_l, r)$ that covers $C$. As $C$ is compact, there exists a positive constant $c$ (depending on the dimension) such that $N_r \leq cr^{-d} \vee 1$, where $d$ is the intrinsic dimension of $C$. Choose $r = \frac{\tau}{2}$ to get,

$$\alpha > 1 + d\, \frac{\log\left(c(\frac{\tau}{2})^{-d} \vee 1\right)}{\log(a_1/a_0)}. \tag{A.4}$$

The first part of the proof finishes by taking $\beta_0 = (c(\frac{\tau}{2})^{-d} \vee 1)$.

If $C$ is smooth enough so that the length of geodesics in $C$ is uniformly bounded by $R$, we can take instead $\gamma_i$ to be the geodesic from $x$ to $y$ to obtain the bound

$$D_\alpha^d(x, y) \leq \int_{\gamma_i} \frac{1}{f^{(\alpha-1)/d}} \leq a_1^{(1-\alpha)/d} R,$$

instead of (A.2). The rest of the proof follows verbatim to obtain

$$\alpha > 1 + d \frac{\log(R/2\tau)}{\log(a_1/a_0)}. \tag{A.5}$$

$\square$

*Proof of Proposition 2.5.* The proof follows the same lines as the one of Proposition 2.2, but we need to be more careful. Recall that $C$ is compact, $f \geq a_1$ in $C$ and $a_0$ is such that there exists a region around each cluster that strictly separates the level-sets $f^{-1}(a_1)$ and $f^{-1}(a_0)$ with the projection being well defined. More precisely, we have that for

$$\eta = \inf \left\{ |s - t| \, : \, s \in f^{-1}([0; a_0]), \, t \in f^{-1}([a_1; \infty]) \right\} > 0,$$

the inequality $\tau = \text{rch}(C) > \eta$ is verified.

We first prove that $\text{rch}(C^\eta) = \text{rch}(C) - \eta = \tau - \eta$. To do that, observe that for any $\delta > 0$, $(C^\eta)^\delta = C^{\eta+\delta}$. In fact, if $x \in (C^\eta)^\delta$, as $C$ is compact, we know that there exists at least one projection of $x$ onto $C$ denoted by $\pi_C(x)$. Applying the customs passage theorem, we have $\{[x, \pi_C(x)] \cap \partial(C^\eta)\} \neq \emptyset$. Call $z$ the point lying at this intersection. We have,

$$d_C(x) = \inf_{p \in C} |x - p| \leq |x - z| + \inf_{p \in C} |z - p| \leq \delta + \eta.$$

Then $x \in C^{\eta+\delta}$. On the other hand, for $x \in C^{\eta+\delta}$ either $d_C(x) \leq \eta$ and $x \in (C^\eta)^\delta$, or $d_C(x) > \eta$. Suppose $d_{C^\eta}(x) > \delta$. As we assumed that $d_{C^\eta}(x) > \delta$, $|x - z| > \delta$ and

$$|x - \pi_C(x)| = |x - z| + |x - \pi_C(x)| > \delta + \eta.$$

A contradiction. Hence $d_{C^\eta}(x) \leq \delta$ and $x \in C^{\eta+\delta} = (C^\eta)^\delta$. This proves our claim

$$\text{rch}(C^\eta) = \text{rch}(C) - \eta = \tau - \eta.$$

The bound from below for the macroscopic Fermat distance between two points lying in two different clusters is the same as in A.1, and we omit it here.

The upper bound for the macroscopic Fermat distance between two points lying in the same cluster is also very similar. The only difference is that we need to build an $r$-covering of $C^\eta$ instead of $C$, with $r < \text{rch}(C^\eta) = \tau - \eta$. We define the shortest path (in the neighborhood graph restricted to $C_i^\eta$) inside $(C_i^\eta)^r = C_i^{\eta+r}$, $1 \leq i \leq m$, in the same manner. The length of this path is smaller than $2(\tau - \eta)N_r$, where $N_r$ is the covering number of $C^\eta$.

We first project this path on $C^\eta$, which will be projected on $C_i^\eta$ by construction. We project this new path on $C$, which is still by construction projected on $C_i$. Applying two times Federer's theorem ((8), Theorem 4.8-(8)), the condition A.3 on $\alpha$ becomes

$$a_0^{(1-\alpha)/d} 2\tau > a_1^{(1-\alpha)/d} \left( \frac{\tau - \eta}{\tau - \eta - r} \right) \left( \frac{\tau}{\tau - \eta} \right) N_r \, 2r,$$

which is equivalent to

$$\alpha > 1 + d \frac{\log \left( \frac{r}{\tau - (r+\eta)} N_r \right)}{\log(a_1/a_0)}.$$

$\square$

Next, we prove Proposition 3.2. We will use the following lemma, whose proof is elementary and is omitted.

**Lemma A.1.** *Assume $\mathcal{M}$ is compact and that for all $x \in \mathcal{M}$, $|f(x)| > \iota > 0$. Then for every $\epsilon > 0$ there is $\delta > 0$ such that for every $u, v \in \mathcal{M}$ with $|u - v| \leq \delta$ we have $|D_\alpha^d(u, x) - D_\alpha^d(v, x)| \leq \epsilon$ for every $x \in \mathcal{M}$.*

*Proof of Proposition 3.2.* We first prove that the definition of $\alpha$-clusters implies with overwhelming probability for $n$ large enough. Using the definition of $\alpha_0$, for $\alpha > \alpha_0 > 1$ there exists $\epsilon > 0$ such that,

$$D_\alpha^d(x, y) \leq D_\alpha^d(x, y') - \epsilon, \text{ for all } x, y \in C_i, \, y' \in C_j, \, j \neq i. \tag{A.6}$$

Consider the events,

$$A_{n,\alpha} = \left\{ \left| n^{(\alpha-1)/d} D_{Q_n,\alpha}(x, y) - \mu D_\alpha^d(x, y) \right| \leq \epsilon/3, \text{ for all } x \in Q_n \right\}.$$

By Proposition 2.6 in(9), there is $\gamma > 0$ and $c > 0$ such that for $n$ large enough

$$P(A_{n,\alpha}^c) \leq e^{-cn^\gamma} \quad \forall 1 \leq i \leq m.$$

Now, on $A_{n,\alpha}$, we have for $x, y \in C_i$ and $y' \in C_j$, $j \neq i$,

$$\begin{aligned}
n^{(\alpha-1)/d} D_{Q_n,\alpha}(x, y) &\leq \mu D_\alpha^d(x, y) + \epsilon/3 \\
&\leq \mu D_\alpha^d(x, y') + \epsilon/3 - \epsilon \qquad \text{(by the clustering condition (2.1))} \\
&\leq n^{(\alpha-1)/d} D_{Q_n,\alpha}(x, y') + 2\epsilon/3 - \epsilon.
\end{aligned}$$

Since $2\epsilon/3 - \epsilon = -\epsilon/3 < 0$, we have

$$n^{(\alpha-1)/d} D_{Q_n,\alpha}(x, y) \leq n^{(\alpha-1)/d} D_{Q_n,\alpha}(x, y') - \epsilon/3,$$

on $A_{n,\alpha}$. Hence

$$P(F(\alpha, n)^c) \leq \mathbb{P}(A_{n,\alpha}^c) \leq e^{-cn^\gamma},$$

for $n$ large enough.

Next, take $\alpha < \alpha_0$ such that $(\mathcal{C}, f)$ is not $\alpha-$feasible and suppose that the microscopic clustering conditions (3.1) are satisfied, *i.e.*, there exists $\epsilon > 0$ such that for every $i \neq j$, $x, y \in C_i \cap Q_n$ and $y' \in C_j \cap Q_n$ we have

$$n^{(\alpha-1)/d} D_{Q_n,\alpha}(x, y) \leq n^{(\alpha-1)/d} D_{Q_n,\alpha}(x, y') - \epsilon. \tag{A.7}$$

So, on $A_{n,\alpha}$ we have,

$$\begin{aligned}
\mu D_\alpha^d(x, y) &\leq n^{(\alpha-1)/d} D_{Q_n,\alpha}(x, y) + \epsilon/3 \\
&\leq n^{(\alpha-1)/d} D_{Q_n,\alpha}(x, y') + \epsilon/3 - \epsilon \qquad \text{(by (A.7))} \\
&\leq \mu D_\alpha^d(x, y') + 2\epsilon/3 - \epsilon.
\end{aligned}$$

We have,

$$D_\alpha^d(x, y) \leq D_\alpha^d(x, y') - \epsilon/3\mu,$$

which can not hold since $(\mathcal{C}, f)$ is not $\alpha-$feasible. This means that $A_n \subset F(n, \alpha)^c$ and the conclusion of the proposition follows. $\square$

*Proof of proposition 4.1.* The study of the statistic $D_{Q_n,\alpha}^{d=1}$ boils down to the study of the uniform spacings given by $\Delta_i = q^{(i+1)} - q^{(i)}$ for $i = 1, 2, \ldots, n-1$, $\Delta_0 = q^{(1)}$ and $\Delta_n = 1 - q^{(n)}$. Using that the joint density $p$ of the order statistics $q^{(1)}, q^{(2)}, \ldots, q^{(n)}$ is given by

$$p(t_1, t_2, \ldots, t_n) = n! \, \mathbb{1}_{\{0 \leq t_1 \leq \ldots \leq t_n\} \leq 1\}}$$

it is easy to derive the joint distribution of the vector $(\Delta_0, \ldots, \Delta_{n-1})$ which is uniformly distributed on the $n$ dimensional simplex denoted by

$$\mathcal{S}_n = \left\{ (s_0, s_1, \ldots, s_{n-1}) \in \mathbb{R}^n, \; s_0, s_1, \ldots, s_{n-1} > 0, \; \sum_{i=0}^{n-1} s_i \leq 1 \right\}.$$

That is $(\Delta_0, \ldots, \Delta_n) \sim \text{Dir}(a)$, where $\text{Dir}(a)$ denotes the flat Dirichlet distribution with parameter $a \in \mathbb{R}^{n+1}$, $a = (1, \ldots, 1)$ and $\Delta_n = 1 - \sum_{i=0}^{n-1} \Delta_i$. The moments of the Dirichlet distribution can be easily found as

$$\mathbb{E}[\Delta_i^\alpha] = \mathbb{E}_{(\Delta_0, \ldots, \Delta_n) \sim \text{Dir}((1,1,\ldots,1))}[\Delta_0^\alpha] = \frac{\Gamma(n+1)}{\Gamma(n+\alpha+1)} \Gamma(\alpha+1),$$

$$\mathbb{E}[\Delta_i^\alpha \Delta_j^\alpha] = \mathbb{E}_{(\Delta_0, \ldots, \Delta_n) \sim \text{Dir}((1,1,\ldots,1))}[\Delta_0^\alpha \Delta_1^\alpha] = \frac{\Gamma(n+1)}{\Gamma(n+2\alpha+1)} \Gamma(\alpha+1)^2.$$

The relative asymptotic behaviour of Gamma functions given by (7)

$$\frac{\Gamma(x+a)}{\Gamma(x+b)} = x^{a-b} \left( 1 + \frac{(a-b)(a+b-1)}{2x} + \mathcal{O}\left(\frac{1}{x^2}\right) \right). \tag{A.8}$$

implies that $\Gamma(n+\alpha+1)/(n+1)^\alpha \Gamma(n+1) \to 1$ as $n \to \infty$. Then,

$$\lim_{n \to \infty} n^\alpha \mathbb{E}[\Delta_i^\alpha] = \Gamma(\alpha+1), \qquad \lim_{n \to \infty} n^{2\alpha} \mathbb{E}[\Delta_i^\alpha \Delta_j^\alpha] = \Gamma(\alpha+1)^2.$$

Finally,

$$n^{\alpha-1} \mathbb{E}[D_{Q_n,\alpha}^{d=1}] = n^{\alpha-1} \mathbb{E}\left[ \sum_{i=0}^n \Delta_i^\alpha \right] = \mathbb{E}\left[ (n\Delta_1)^\alpha \right] \xrightarrow{n \to \infty} \Gamma(\alpha+1).$$

On the other hand, the second moment can be computed as

$$\mathbb{E}[(D_{Q_n,\alpha}^{d=1})^2] = \mathbb{E}\left[ (\sum_{i=0}^n \Delta_i^\alpha)^2 \right] = (n+1)\mathbb{E}\left[ \Delta_1^{2\alpha} \right] + n(n+1)\mathbb{E}[\Delta_1^\alpha \Delta_2^\alpha]$$

$$= (n+1)\left( \mathbb{E}\left[ \Delta_1^{2\alpha} \right] - \mathbb{E}[\Delta_1^\alpha]^2 \right) + \mathbb{E}[D_{Q_n,\alpha}^{d=1}]^2$$

$$+ n(n+1)\left( \mathbb{E}[\Delta_1^\alpha \Delta_2^\alpha] - \mathbb{E}[\Delta_1^\alpha]\mathbb{E}[\Delta_2^\alpha] \right).$$

Rearranging the terms in the last expression, we obtain

$$n \, \text{Var}[n^{\alpha-1} D_{Q_n,\alpha}^{d=1}] = \frac{n+1}{n} \left( \mathbb{E}[n^{2\alpha} \Delta_1^{2\alpha}] - \mathbb{E}[n^\alpha \Delta_1^\alpha]^2 \right)$$

$$+ (n+1) \, \text{Cov}(n^\alpha \Delta_1^\alpha, n^\alpha \Delta_2^2).$$

As $n \to \infty$, the first term on the right-hand side converges to $\Gamma(2\alpha+1) - \Gamma(\alpha+1)^2$. Based again in Equation (A.8), we have

$$\mathbb{E}[\Delta_1^\alpha \Delta_2^\alpha] - \mathbb{E}[\Delta_1^\alpha]\mathbb{E}[\Delta_2^\alpha] = \left( \frac{\Gamma(n+1)}{\Gamma(n+2\alpha+1)} - \frac{\Gamma(n+1)^2}{\Gamma(n+\alpha+1)^2} \right) \Gamma(\alpha+1)^2$$

$$= \left( \frac{1}{n_+^{2\alpha}} \left( 1 - \frac{2\alpha(2\alpha-1)}{2n_+} + \mathcal{O}(n^{-2}) \right) \right.$$

$$\left. - \frac{1}{n_+^{2\alpha}} \left( 1 - \frac{\alpha(\alpha-1)}{2n_+} + \mathcal{O}(n^{-2}) \right)^2 \right) \Gamma(\alpha+1)^2$$

$$= \left( -\frac{\alpha^2}{n_+} + \mathcal{O}(n^{-2}) \right) \frac{\Gamma(\alpha+1)^2}{n_+^{2\alpha}},$$

where $n_+ = n+1$. This implies

$$\lim_{n \to \infty} (n+1) \, \text{Cov}(n^\alpha \Delta_1^\alpha, n^\alpha \Delta_2^2) = -\alpha^2 \Gamma(\alpha+1)^2,$$

which finally gives

$$\lim_{n\to\to\infty} n \operatorname{Var}[n^{\alpha-1} D_{Q_n,\alpha}^{d=1}] = \Gamma(2\alpha+1) - (\alpha^2+1)\Gamma(\alpha+1)^2.$$

$\square$

*Proof of Proposition 4.2.* Note that we can turn results for a Poisson process of intensity $n$ in the hypercube to a scaled Poisson process $n^{1/d}\mathcal{Q}_n$ which is essentially (up to exponentially small probability events) to a unit Poisson process $\mathcal{P}$ in the hyperplane (see Theorem 2.4 in (12)). Hence with $l = n^{1/d}$, we replace $n^{(\alpha-1)/d} D_{\mathcal{Q}_n,\alpha}^d(0, e_1)$ by $l^{-1} D_{\mathcal{P},\alpha}^d(0, le_1)$, and now work with a unit Poisson process. The proof is based on constructing a specific (sub-optimal) path that has been considered previously in (13). Define the sets,

$$\mathcal{D}_t(q) = \{q + b \in \mathbb{R}^d, \ \le b_1, \ \text{for } 2 \le i \le d\} 0 \le b_1 \le t, \ 0 \le \sigma_i b_i$$

where $\sigma_i = -1_{q_i > 0} + 1_{q_i \ge 0}$. Let us now construct inductively the specific path of $l$ points $q_0, \ldots q_l$ such that for $k \le l$,

$$\tilde{q}_0 = 0,$$
$$X_k = \inf\{t > 0, \ \text{there is a point in } \mathcal{D}_t(\tilde{q}_{k-1})\},$$
$$\tilde{q}_k = \text{particle present in } \mathcal{D}_{X_k}(\tilde{q}_{k-1}),$$

Finally, $q_{l+1} = le_1$. Observe that,

$$|\tilde{q}_k - \tilde{q}_{k-1}|^\alpha \le d^{\frac{\alpha}{2}} X_k^\alpha \text{ and } |\tilde{q}_{l+1} - le_1|^\alpha \le \sum_{i=1}^l d^{\frac{\alpha}{2}} X_i^\alpha.$$

Using basic properties of Poisson processes, we get that the random variables $(X_i)_{1 \le i \le l}$ are i.i.d. Now,

$$\mathbb{P}(X_i \ge x) = \mathbb{P}(\text{ no points in } \mathcal{D}_x(0)) = e^{-(1/d)x^d}.$$

Therefore,

$$\mathbb{P}(X_i^\alpha \ge x) = e^{-(1/d)x^{d/\alpha}} = e^{-(1/d)x^{1/\theta}}.$$

Hence,

$$D_{\alpha,\mathcal{P}}^d(0, le_1) \le \sum_{i=1}^l |\tilde{q}_i - \tilde{q}_{i-1}|^\alpha + |\tilde{q}_l - le_1|^\alpha,$$

which gives that

$$\mathbb{P}(D_{\alpha,\mathcal{P}}^d(0, le_1) \ge lx) \le \mathbb{P}\left(2 \sum_{i=1}^N |\tilde{q}_i - \tilde{q}_{i-1}|^\alpha > lx\right),$$
$$\le \mathbb{P}\left(2 \sum_{i=1}^l |\tilde{q}_i - \tilde{q}_{i-1}|^\alpha > lx\right)$$
$$\le \mathbb{P}\left(\sum_{i=1}^l X_i^\alpha > lx/2d^{\alpha/2}\right).$$

Note that if $\alpha > d$, we cannot use the usual large deviation bounds as the variable $X_i$ does not have an exponential moment. We can, however, use their usual moments. Our computations are similar to the ones in (28).

Observe that $\mathbb{P}(Y \ge x) = e^{-x^{\frac{1}{\theta}}/d}$ implies $\mathbb{E}(Y^\gamma) = d^{\gamma\theta}\Gamma(\gamma\theta+1)$.

Let $Y_i = X_i^\alpha$ and take $\gamma \geq 1$. Then,

$$\mathbb{P}\left(\sum_{i=1}^l Y_i \geq t\right) \leq E\left[\left(\sum_{i=1}^l Y_i\right)^\gamma\right] t^{-\gamma}, \tag{A.9}$$

$$\leq \left(\sum_{i=1}^l \mathbb{E}(Y_i^\gamma)^{1/\gamma}\right)^\gamma t^{-\gamma}, \tag{A.10}$$

$$= l^\gamma d^{\gamma\theta} \Gamma(\gamma\theta + 1) t^{-\gamma}, \tag{A.11}$$

$$\leq l^\gamma d^{\gamma\theta} (\gamma\theta)^{\gamma\theta} t^{-\gamma}. \tag{A.12}$$

We use the bound $\Gamma(x+1) \leq x^x$ for $x \geq 1$ in the last line. Taking $t$ such that $l^\gamma d^{\theta\gamma} (\gamma\theta)^{\gamma\theta} t^{-\gamma} = e^{-\gamma}$, we can state that for all $\gamma \geq 1$

$$\mathbb{P}\left(\sum_{i=1}^l Y_i \geq e d^\theta (\gamma\theta)^\theta l\right) \leq e^{-\gamma},$$

and $\gamma^\theta c = x$ with $c = 2 e d^{\theta+\alpha/2}\theta^\theta$, leads to

$$\mathbb{P}\left(\frac{1}{l}\sum_{i=1}^l Y_i \geq \frac{x}{2d^{\alpha/2}}\right) \leq e^{-(x/c)^{1/\theta}},$$

for all $x > c$. Hence

$$\mathbb{E}\left(\frac{1}{l}\sum_{i=1}^l Y_i\right)^k \leq c + \int_c^\infty e^{-(x/c)^{1/k\theta}} dx,$$

$$\leq c + c\Gamma(k\theta + 1).$$

We obtain,

$$\mathbb{E}\left(\left(l^{-1}D_{\alpha,\mathcal{P}}^d(0, le_1)\right)^k\right) \leq c + c\Gamma(k\theta + 1).$$

$\square$

