# OpenReview forum: "Choosing the parameter of the Fermat distance: navigating geometry and noise"
_TMLR — Accepted by TMLR_

### Review · Reviewer_oM8B · 2023-12-22

**Summary Of Contributions:**

This paper proposes a theoretical and experimental study of the parameter $\alpha$ in the Fermat distance for the clustering problem. The Fermat distance extends the Euclidean distance, by including information about the underlying manifold on which the data is supported and the density function from which the data points are sampled. The authors highlight the importance of the $\alpha$ parameter for the performance of the clustering problem, and prove the existence of a lower critical value, given by an explicit quantity depending on the geometric complexity of the problem and which delimits its feasibility. Experiments are conducted to empirically confirm that the performance of the clustering problem strongly depends on $\alpha$.

**Audience:**

Yes

**Broader Impact Concerns:**

I don't think that this paper requires a Broader Impact Statement.

**Claims And Evidence:**

Yes

**Requested Changes:**

- It is said (page 2) that the re-scaling of the distance by the power $1/\alpha$ "carries implications for practical implementations", can you develop on this?

- Regarding the experiments in Figure 1, it might be interesting to increase the number of points $n$, to see in the results are more in line with the theoretical value of the critical value (i.e. the clustering remains good for smaller values of $\alpha$).

- Can the theoretical critical value $\alpha_0$ of the simulated experiment in Section 5.2 be computed (at least approximatively)?

- Typo : in the proof of Prop 3.2, I think that in (A.6) you meant to include the parameter $\mu$ in the clustering condition in order to prove the inequality on the sample Fermat distance (page 13)? Moreover, when you say "(by the clustering condition (2.1))", I assume you are referring to (A.6) (same inequality).

I think these comments should be addressed for paper acceptance.

**Strengths And Weaknesses:**

Strengths :
- The authors' objective of optimizing (in terms of performance) the choice of the parameter  $\alpha$ in order to better perform a clustering task is very relevant in my opinion. I particularly appreciated the proposal of theoretical results to support the behaviours shown in experiments, for a clustering problem conducted using K-medoids method.
- A study on the sample Fermat distance for understanding its variability, in particular for large value of $\alpha$, is also conducted in dimension one, and verified in the experiments.

Weaknesses :
- As the authors point out throughout the paper, the theoretical analysis of a critical value for the parameter $\alpha$ in the Fermat distance is (i) application dependant and (ii) very difficult to achieve in general cases. Consequently, the analysis carried out in the paper seems somewhat preliminary.
- The experiments are conducted in the specific case of clustering, although, as mentioned by the authors, a classification task could also be carried out in the same framework.

However, to the best of my knowledge, this is the first attempt at a theoretical analysis of the parameter $\alpha$ in the Fermat distance, which in my opinion paves the way for future research.

---

> ### Author Response · Authors · 2024-03-22
>
> We agree with all the remarks, and we will change the paper accordingly. Some comments:
>
> -   The rescaling of the distance by $1/\alpha$ changes the nature of the numerical problems that may appear for large $\alpha$ (without making them vanish). We however believe that all the results given here would qualitatively (i.e. in terms of phenomenology) translate to this case.
> -    Yes,  the reviewer is right, we could make a more intensive simulation to emphasize the phenomenon better.
> -    Yes, we can compute approximately $\alpha_0$
> -    Yes, we will correct this.

---

### Review · Reviewer_Ck1k · 2023-12-22

**Summary Of Contributions:**

The paper considers the Fermat distance parameterized by $\alpha$, and the clustering problem defined using this distance.

The main results of the paper show that the parameter $\alpha$ needs to be chosen carefully in such a way that the feasibility and variance is balanced. Specifically, the results in Section 2 show that, under certain conditions, a large enough $alpha$ would ensure the clustering problem is feasible. On the other hand, Section 4 shows that with a large $\alpha$ the variability of the sample Fermat distance is very large.

**Audience:**

Yes

**Broader Impact Concerns:**

The work is primarily theoretical and does not immediately induce ethical implications.

**Claims And Evidence:**

Yes

**Requested Changes:**

See the above.

**Strengths And Weaknesses:**

Strength:
- The paper is clearly written with a streamlined story to serve the reader.
- The proved results are interesting, characterizing what would happen for different values of $\alpha$'s for large sample sizes.

Weaknesses:
- Some typos or inaccuracies:
   - Abstract, "exploding": Does it mean exploring?
   - The first paragraph: "it can take any value": To be precise I would write "it can take any integer value in $[1,D]$.
   - Remark 2.4: "we necessary have $a_0=a_1$"
- In Eq. (1.1), I found the definition questionable: I think the minimum is always attained when $k=1$ and it does not make sense to have larger $k$. Am I correct? If so, please fix it.
- The legends of the figures are too small. For example, I can't see the legends of Figure 2 clearly without zooming in.
- I don't see the negative consequences of Proposition 4.1. What can we say about correct recovery or feasibility if the variance is large? Large variance does not seem to have any negative impacts on feasibility. Am I correct?
- There is a huge gap between theory and experiments. Specifically, the theory concerns whether the clustering problem is feasible. The experiments show clustering accuracy of a specific algorithm (say, a Fermat-distance variant of K-medoids). The issue is that such an algorithm does not necessarily solve the clustering problem optimally (when it is feasible), and it rather relies on a good initialization. Note that in Figure 2 the red curve (best performance among different trials) is almost an increasing function of $\alpha$. I would argue that a large $\alpha$ benefits as it makes the clustering problem feasible and furthermore a K-mediods variant with multiple initializations can find a good solution (red curve). This is different from what the paper argues: A large $\alpha$ can be harmful due to the variability. Please justify.

---

> ### Author Response · Authors · 2024-03-22
> **Theoretical and practical tradeoff for the critical parameter**
>
> -  Thanks for the comments. We shall correct the typos and improve the legends' readability.
>
> -   We think that the definition (1.1) is correct. The referee is right in that the infimum is attained (but it is in general attained in more than 1 step). We can change inf to min. The parameter $k$ represents here the number of steps contained in a path. The referee claims that the inf is attained when $k=1$, which is true when $\alpha=1$, but when $\alpha>$, shorter edges are encouraged.
>
> -    Proposition 4.1 and its consequence (4.3) shows that as $\alpha$ gets large, the estimator of the Fermat distance becomes less and less reliable, being more and more noisy. This phenomenon depends in an intricate way both on the dimension and on the geometric properties of the density. In the simulations, the negative consequences brought by a large variance are clearly grasped for instance in Fig 2 and 3.  As a general fact, large variance cannot be good news for estimation. In our case, large variance implies that we cannot rely on the feasibility of the population problem to guarantee the feasibility of the empirical one.
>
> Note that if $\alpha$ gets too large for a fixed number of points $n$, the large variance effect will dominate. The exponential bounds in $\alpha$ on the variance suggest that $n$ should scale exponentially in $\alpha/d$ to achieve a given error tolerance in the estimation.
>
> To clarify this crucial point, we will add a paragraph, explaining the phenomenon in more detail (see also the response to Reviewer XVgj).

---

### Review · Reviewer_XVgj · 2024-03-08

**Summary Of Contributions:**

The paper explores the theoretical implications of employing both statistical and empirical versions of Fermat distance in clustering problems. Fermat distance between any two points $x,y\in\mathcal{M}\subseteq\mathbb{R}^D$ (where $\mathcal{M}$ denotes a $d$-dimensional Reimannian manifold with $d\leq D$) and with respect to a probability density $f$ over the manifold is defined, in the statistical sense, as:

$D^d_{\alpha}=\inf_{\gamma\in\Gamma_{x,y}}\int\frac{1}{f^{(\alpha-1)/d}}$,

where $\inf$ is taken w.r.t. all integrable paths over $\mathcal{M}$ connecting $x$ to $y$, i.e., $\Gamma_{x,y}$. The empirical version tries to approximate the above formula via $n$ randomly sampled points from $f$, which turns the above continuous minization problem into a tractable combinatorial optimization problem (Eq. (1.1) of the paper). Uniform convergence of the empirical version to the true statistical one is already proved in the literature.

**Claims**:

The authors claim and subsequently prove that for sufficiently large values of $\alpha$, the metric, even in the empirical scenario with a sufficiently large sample size $n$, becomes valuable for clustering specific statistical mixture models or well-separated high-density regions of a single density function $f$.

**Contributions**

- In Proposition 2.2, the authors establish the validity of their claim for the asymptotic case ($n\rightarrow\infty$) using various geometric tools. Proposition 3.2 extends this validation to the non-asymptotic scenario ($n<\infty)$. However, reservations are expressed in the Weaknesses section.

- The authors illustrate their findings through a toy example in Section 3.1 with a piecewise-constant density function over a hyper-cube containing two distinct inner rings. Section 4.1 addresses the drawbacks of choosing large $\alpha$, proposing the existence of an optimal "window." The authors substantiate this claim with a one-dimensional toy example, showcasing the diminishing ratio of "std-to-mean" for Fermat distance as $n^{-1/2}$ but exponential growth concerning $\alpha." Section 4.2 explores the higher-dimensional case again in a limited setting.

**Experiments**

Paper also presents a number of numerical experiments, but I didn't manage to check them entirely.

------------------

**Overal Rating**

This a nice theory paper and is enjoyable to read. Mathematical derivations are crisp and valid, as far as I have checked (I have also checked some of the proofs, but not all of them). The mathematical derivations and proofs are sound, but concerns arise regarding the overall impact. None of the statements are exceptionally strong, applications seem constrained, and some contributions rely on toy examples rather than rigorous mathematical guarantees.

My current vote is weak reject, but I am open to reconsider this rating it if authors manage to address my concerns properly. Also, I would like to see assessments from other reviewers as well.

**Audience:**

Yes

**Claims And Evidence:**

Yes

**Requested Changes:**

Please see the weaknesses section.

**Strengths And Weaknesses:**

**Strengths**:

- Paper is well-written and addresses an interesting issue within the realm of machine learning, particularly in the context of clustering.
- The mathematical derivations appear robust and devoid of any significant mathematical errors.

- The paper follows an interesting trajectory, incorporating a variety of tools, notably from high-dimensional geometry and analysis. However, it's worth noting that the current strength of the claims may not be particularly interesting at this point.

**Weaknesses**:

- It would enhance the paper's practical applicability if the authors delved more extensively into the potential applications of Fermat distance in real-world settings. As it stands, the paper appears more focused on the mathematical aspects.

- The definition of "clusters" in Definition 2.1 appears overly restrictive. Specifically, the assumption of the existence of a metric (Fermat distance in this case) where all inter-cluster distances are consistently smaller than intra-cluster distances seems applicable to only a limited subset of clustering scenarios, notably a special case of "well-separated" ones.

- Due to the stringent assumptions, the proof of Proposition 2.2 seems somewhat simplistic, potentially hampering the overall contribution of the paper. Nonetheless, the incorporation of mathematical tools such as the "reach of set" or leveraging Theorem 4.8 from [6] is indeed a commendable contribution.

- **Major concern 1**: The conditions outlined in Propositions 2.2 and 3.2 are posited as "sufficient" for identifying optimal clusters $\left(C_i\right)_{i=1}^{m}$. While it's evident that they are "necessary," establishing their sufficiency requires an algorithm with a rigorously proven capacity to identify clusters (computational complexity aside). I am not convinced the conditions in Propositions 2.2 and 3.2 are "sufficient" for finding the optimal (true) clusters.

- **Major concern 2**: Proposition 3.2 appears hastily presented and somewhat vague. The mere "existence" of constants $\gamma, \delta, c, \bar{n}$ lacks depth; they should be properly bounded as functions of other parameters. For instance, there should be clear bounds on $\bar{n}$ to prevent divergence or restrictions on the values of $\gamma$, which might vary significantly. In other words, $\bar{n}$ might diverge to infinity in some cases, or $\gamma$ could become very small or very large (the latter is not good for the converse statement).

- **Major concern 3**: The overal contribution of this work is not entirely clear. While the authors establish general statements regarding the clusterability of certain mixture models via the utilization of Fermat distance with a propely chosen $\alpha$, the inherent complexity of these problems appears to have transitioned into abstract and possibly intractable quantities, such as "the reach of the cluster sets." This shift raises concerns about the practical utility of Fermat distances in real-world applications. In other words, I am afraid authors have turned one form of complexity into another form, which is still not tractable and thus undermines the use of Fermat distances for practical use.

- The analyses in Section 3.1, 4.1, and 4.2 rely on carefully selected toy examples rather than providing broad mathematical guarantees. Admittedly, offering general assurances in this field might be challenging and could correspond to open problems in mathematics.

----------

**Minor comments**:

- Abstract: exploding or exploiting?

- The abstract lacks clarity and might require a major revision. Notably, Fermat distance remains undefined, yet the discussion on $\alpha$ has already started.

- The minimization problem in Eq (1.1) of Definition 1.1 appears mathematically flawed. The objective for $\inf$ is ambiguous, seemingly involving the choice and ordering of $q_i$s, but lacks precise clarification.

- In Remark 2.4, the authors assert that the conditions in Proposition 2.2 necessitate $f$ to be discontinuous. However, in my understanding $f$ can be continuous and conditions in Propostion 2.2 still hold, for example by forcing $C_i$s to be distanced away from each other, right?

---

> ### Author Response · Authors · 2024-03-22
> **Potential threats against the method**
>
> - We will include a paragraph on the use of Fermat distance in real-world applications. It has already been shown to be useful in topological data analysis to detect cancer fingerprints (Carpio et al. Fingerprints of cancer by persistent homology. bioRxiv, 2019, p. 777169.), in signal analysis to detect periodicity and anomalies in ECG and in the reconstruction of dynamical systems to mention some of them (Fernández et al (2023). Intrinsic persistent homology via density-based metric learning. Journal of Machine Learning Research, 24(75), 1-42.).
>
> - We remark that our definition of clusters is less restrictive than the usual one based on sub-level sets of the density: clusters defined as sub-level sets of the density verify to be also clusters according to our definition. We mention this p.2, just before 2.1.
>
> - Major concern 1:
>
> The primary focus of our paper is to explore the potential use of Fermat distance within machine learning frameworks in terms of $\alpha$ values. Focusing on clustering, the premise taken is that if the clusters are well separated in terms of Fermat distance, a very simple algorithm like k-means or more generally an EM algorithm will be indeed sufficient. We agree with the reviewer that a formal proof giving guarantees for fixed $n$ (based on several assumptions on the data) could be given, but we thought it was outside the scope of the paper. We could nevertheless include it in a next version.
> Proposition 2.2 demonstrates a "phase transition" in $\alpha$, indicating that clustering becomes "exactly feasible" in the large $n$ limit only for sufficiently large values of alpha. Its main significance lies in the quantification of this threshold, which directly reflects three key aspects: the dimensionality (d), the geometrical difficulty of cluster separation, and regularity assumptions on clusters. Specifically, the dependency on dimension is linear for "regular enough" problems and quadratic otherwise, with regularity expressed in terms of reach. These findings have implications for the success of clustering algorithms based on Fermat distance, particularly regarding the influence of noise. Remark 2.3 underscores the importance of these points.
>
>
> - Major concern 2:
>
> Thanks for pointing this out. We can indeed be more precise when formulating Proposition 3.2. The constant $\gamma$ can be easily bounded as a function of the dimension and $\alpha$.
> It might be much more difficult to get useful estimates for $\bar n$, but that can be considered and in any case, commented.
>
> - Major concern 3:
>
> We believe that our findings convey important messages, which may not have been adequately emphasized in the paper.
>
> Takeaway 1:
>  When clusters exhibit regularity (e.g., uniformly bounded diameters in geodesic distance), the parameter $\alpha_0$ exhibits linear dependence on dimensionality (d), and we proved that the second moment of the distance remains bounded, thereby ensuring the feasibility of the clustering task even in high dimensions with a reasonable number of data points.
>
> Takeaway 2:
> In cases where clusters lack regularity, $\alpha_0$ demonstrates quadratic dependence on dimensionality. We conjectured that this implies an exponential dependence on the dimension for the variance of the Fermat distance. Consequently, the practical feasibility of the clustering task in high dimensions becomes too costly, as it would need an exponential scaling of the number of points with dimensionality.
> We can underline this more clearly in a new revision.
>
> We agree with the referee that in practical problems we don't know a priori if the clusters are regular or not. Still, we think it is always useful to understand what are the potential threats against the methods we are using to try to solve a specific practical problem.
>
> - The analyses in Section 3.1, 4.1, and 4.2 rely on carefully selected toy examples rather than providing broad mathematical guarantees. Admittedly, offering general assurances in this field might be challenging and could correspond to open problems in mathematics.
>
> - Concerning the last remark, those examples indeed just illustrate some of the theoretical points studied but do not have the pretension to be generic. As the referee points out, providing rigorous mathematical proofs of this fact is closely related to well-known problems in mathematics that have been open for more than 50 years. We remark that we also present experiments with the MNIST dataset.

---

### Decision · Action_Editor_FqqL · 2024-04-24

**Recommendation:** Accept as is

**Comment:**

Based on the reviews, the paper makes significant theoretical contributions by exploring the Fermat distance in clustering problems. The detailed mathematical proofs and the introduction of new propositions about the Fermat distance’s behavior in clustering underscore its potential use in ml problems..

Some reviewers have noted that the application of these theoretical insights into practical scenarios is limited and somewhat abstract. This is sometimes expected with such theoretical approaches. The reliance on toy examples and the lack of broad empirical validations are seen as potential weaknesses. Despite these concerns, which have been discussed during the rebutal, the focus on Fermat distance and its detailed theoretical examination provides a case for its publication, as the field is relatively unexplored and the paper opens new directions for future research.

In general, reviewer feedback is positive, all leaning towards acceptance, despite reservations about the paper's practical implications and the robustness of some claims.

**Audience:**

The paper would definilty interest a subset of TMLR's audience, particularly those engaged in geometry and clustering. The exploration of Fermat distance in this contex  presents new interesting insights into  this area.

**Claims And Evidence:**

All referees agree that the submission's claims are generally supported by the evidence provided.  While reviewer XVgj points out that some claims are substantiated primarily through toy examples rather than rigorous mathematical proofs, the theoretical contributions, for i instance Prop. 2.2 & 3.2, are deemed enough to the claims regarding the uttility of Fermat distance in (some) clustering problemss. This is further supported by numerical experiments, although there seems to be  a consensus that further empirical study could strengthen the paper's claims.